# Clinical Impact of Single Nucleotide Polymorphism in CD-19 on Treatment Outcome in FMC63-CAR-T Cell Therapy

**DOI:** 10.3390/cancers15113058

**Published:** 2023-06-05

**Authors:** Katja Seipel, Mariesol Abbühl, Ulrike Bacher, Henning Nilius, Michael Daskalakis, Thomas Pabst

**Affiliations:** 1Department for Biomedical Research, University of Bern, 3008 Bern, Switzerland; 2Department of Medical Oncology, Inselspital, Bern University Hospital, 3010 Bern, Switzerland; 3Department of Hematology, Inselspital, Bern University Hospital, 3010 Bern, Switzerland; 4Department of Clinical Chemistry, Inselspital, Bern University Hospital, 3010 Bern, Switzerland

**Keywords:** B-lymphocyte antigen CD19, single nucleotide polymorphism (SNP), minor allele frequency (MAF), CAR-T cell therapy, FMC63-chimeric antigen receptor (FMC63-CAR), Tisagenlecleucel (Kymriah©), Axicaptagene ciloleucel (Yescarta©), Lisocabtagene maraleucel (Breyanzi©)

## Abstract

**Simple Summary:**

FMC63-CAR-T cell therapy, an immunotherapy against CD19 protein expressed on malignant B cells, is effective in diffuse large B-cell lymphoma (r/r DLBCL) with complete response in half of the treated patients. Remarkably, there are two germline CD19 antigen variants due to the single nucleotide polymorphism rs2904880 encoding leucine or valine at amino acid position 174 of the CD19 antigen, which is present in half of the global population. We present evidence that this single nucleotide polymorphism in CD19 has a clinical impact on the treatment outcome in FMC63-anti-CD19-CAR-T cell therapy.

**Abstract:**

Chimeric antigen receptor (CAR)-T cell therapy is effective in patients with relapsed or refractory diffuse large B-cell lymphoma (r/r DLBCL) with response rates of 63–84% and complete response observed in 43–54%. Common germline variants of the target antigen CD19 may elicit different responses to CAR-T cell therapy. The CD19 gene single nucleotide polymorphism rs2904880 encoding leucine or valine at amino acid position 174 of the CD19 antigen was prevalent in 51% of the studied DLBCL patients. In a retrospective comparative analysis of clinical outcome, there were significant differences in CD19 L174 versus V174 carriers: the median time of progression-free survival was 22 vs. 6 months (*p* = 0.06), overall survival was 37 vs. 8 months (*p* = 0.11), complete response rates were 51% vs. 30% (*p* = 0.05), and refractory disease rates were 14% vs. 32% (*p* = 0.04). The single nucleotide polymorphism in CD19 was shown to influence the treatment outcome in FMC63-anti-CD19-CAR-T cell therapy, and the CD19 minor allele L174 predicted a favorable treatment outcome.

## 1. Introduction

Chimeric antigen receptor (CAR) T-cell therapies are approved for patients with relapsed or refractory diffuse large B-cell lymphoma (r/r DLBCL). Patients receiving CAR-T products, including Axicabtagene ciloleucel (Axi-cel) and Tisagenlecleucel (Tisa-cel), have overall response rates of 63–84%, complete response rates of 43–54%, and superior outcomes compared to standard second or later line therapy consisting of salvage chemotherapy followed by autologous stem cell transplantation (ASCT) or other salvage options [1,2,3,4,5,6]. Despite impressive complete response rates, a significant number of CAR-T-treated patients will eventually relapse or fail to achieve complete remission [7,8]. A multicenter clinical trial with patients receiving Axi-cel reported 28% of the patients with primary refractory disease and 30% of the patients relapsed after a median follow-up of 12.8 months [8].

Even though some clinical features may be associated with differing outcomes in response to CAR-T cell treatment [9,10,11], the mechanisms of resistance or relapse are not fully elucidated [12]. Genetic variants of the CD19 target antigen may contribute to the different responses to CAR-T cell treatment. CD19 is a surface marker exclusively expressed on normal and malignant B cells and on B-lineage progenitors, but not on multi-lineage, myeloid, erythroid, or megakaryocyte progenitor cells [13]. Due to this cell type-specific expression, the CD19 antigen has emerged as a preferred target for specific immunotherapy of B-cell malignancies [14,15]. As in other types of cancer cells, the target can be modified in order to evade cytotoxicity induced by the immune system of the body or by immunologic therapies [16]. In fact, relapse after CAR-T cell therapy was reported as due to loss of antigen (CD19−) or despite retention of antigen (CD19+). CD19-negative relapses appear to be relevant in pediatric patients with B-ALL. Clinical reports detected negative CD19 status in 65–75% of all relapses after preceding response to CAR-T cell therapy [17,18]. Studies on the genome of CD19 in pediatric patients with relapsed B-ALL after CAR-T identified multiple somatic mutations enabling skipping or alternative splicing of exon 2, leading to retention or alternate folding of the membranous fraction, thus indicating loss of the CD19 epitope required for CAR-T cells to trigger targeted cell death [19,20,21,22].

The relevance of CD19 epitope loss in relapse after CAR-T cell treatment in DLBCL adult patients is less evident. Clinical trials indicate a rate of only 20–25% of CD19-negative status among patients with progressive disease at any time after CAR-T cell therapy [8,23]. These observations suggest that determining the CD19 status in DLBCL per se is not sufficient to explain the majority of escape mechanisms and alternative mechanisms that may be involved [24,25]. A somatic point mutation p.174.L-V of the CD19 antigen was recently described in lymphoma cells of two DLBCL patients relapsing after CAR-T cell therapy. In vivo, lymphoma cells with the CD19 V174 variant were reported to be non-responsive to FMC63-CAR19-T cell therapy. In vitro, cells with the variant CD19 V174 exhibited inferior response to CAR-T cell-induced cytotoxicity [26]. Notably, FMC63 is the single-chain variable fragment (scFv) of the anti-CD19 antibody present in all FDA-submitted CTL019 products registered for treatment of r/r DLBCL, including Tisagenlecleucel (Kymriah©), Axicaptagene ciloleucel (Yescarta©), and Lisocabtagene maraleucel (Breyanzi©, JCAR017) [27].

Remarkably, there are two germline CD19 antigen variants due to the single nucleotide polymorphism rs2904880 encoding leucine or valine at amino acid position 174 of the CD19 antigen. Genotype frequencies for the CD19 antigen are L/L = 3.4%, L/V = 50%, and V/V = 46.6% [28], indicating that half of the global population carries the minor CD19 L174 variant. In this analysis, we investigated the prevalence of the CD19 L174 allele and clinical outcome in a retrospective study of a B-cell lymphoma cohort treated with FMC63-anti-CD19-CAR-T cell therapy.

## 2. Materials and Methods

### 2.1. Patients

The single-center retrospective observational study was conducted at Bern University Hospital, Berne, Switzerland, in patients treated between 9 January 2019 and 20 March 2023. All patients were followed for at least six months after CAR-T cell therapy. Eighty-eight patients diagnosed with r/r DLBCL were included. Clinical data were collected including clinical characteristics, treatment lines prior to CAR-T cell therapy, response to CAR-T cell therapy, eventual progression of the disease, as well as laboratory parameters before and after CAR-T cell therapy. The study was approved by the ethics committee in Berne, Switzerland (decision number 2022-00203 on 4 May 2022).

### 2.2. Study Endpoints

The primary endpoint of the study was clinical outcome (relapse rate, PFS, and OS) in patients with vs. without CD19 rs2904880. Secondary endpoints included correlation of CAR-T cell persistence, administered CAR-T cell product (Kymriah^®^ vs. Yescarta^®^), as well as correlation with toxicities. CRS and ICANS clinical assessment and grading were performed following the American Society for Transplantation and Cellular Therapy (ASTCT) consensus grading.

### 2.3. Monitoring of CAR-T Cell Kinetics

CAR-T cell construct kinetics were monitored in PB using a previously established ddPCR assay [11,29]. This assay enabled the circulating copies of the intracellular junction domain located between the effector and co-stimulatory domains of the CAR to be quantified. The number of CAR-T copies per μg of haploid DNA was calculated based on measurement of the ribonuclease P protein subunit 30 (RPP30) concentration, which enabled the number of haploid genomes to be estimated. The limit of detection was 20 copies/μg of DNA. Patients with a CAR-T construct concentration below this detection limit were considered as negative.

### 2.4. Statistical Analysis

Progression-free survival (PFS) was defined as time from CAR-T cell infusion to any of the following events: relapse, death, or lost to follow-up. Overall survival (OS) was defined as time from CAR-T cell infusion to date of death. GraphPad Prism 8^®^ was used to create the graphical representations in the figures and statistical analyses for the tables and figures. PFS and OS curves were analyzed using the Kaplan–Meier method and Mantle–Cox test. Categorical data in tables were analyzed using Fisher’s exact test, whereas the unpaired t-test was used to evaluate parametric data and the Mann–Whitney U-test was used to analyze non-parametric data. Multivariate Cox proportional hazards models were fitted to the data and hazard ratios were calculated using the “survival” package in R version 4.1.2 (2021). The following variables were included in the statistical analysis: age, sex, DLBCL transformation, initial lymphoma stage, number of previous therapy lines, previous stem cell transplantation, bridging therapy, CAR-T cell product, remission status at CAR-T cell infusion, LDH level before CAR-T cell therapy, CRS, and ICANS. P-values below 0.05 were considered statistically significant and percentage results were rounded to whole numbers.

### 2.5. CD19 Gene Analysis

Genomic DNA was extracted from PBMCs isolated from peripheral blood collected before CAR-T cell infusion. DNA fragments were amplified using FIREPol (Solis Biodyne, Tartu, Estonia) and the following gene-specific primers covering exons 3 and 4 of the CD19 gene: forward primer 5′-CTCCCTCTCCTGGGTGTCTCTGCA-3′, and reverse primer 5′-CCCAGTACC-CCCACAGATGCCT-3′. Sanger sequencing was performed at Microsynth, Balgach, Switzerland.

## 3. Results

### 3.1. Prevalence of the CD19 L174 Allele

The sequence of the CD19 gene spanning exons 3 and 4 was determined in the peripheral blood of 120 patients evaluated for CAR-T cell therapy at our center. We found that 59 patients (49.2%) carried two alleles encoding CD19 V174. A total of 56 patients (46.7%) had one allele with the single nucleotide polymorphism rs2904880 encoding L174, and 5 patients (4.2%) carried two alleles with SNP rs2904880, indicating a minor allele frequency of 0.2–0.3. The observed CD19 allele frequencies coincided with those expected in the general population with a global minor allele frequency of 0.23 (TOPMED) to 0.29 (ALFA) [30,31].

### 3.2. Baseline Clinical Characteristics of the DLBCL Patient Cohort

The patient cohort ultimately receiving FMC63-anti-CD19-CAR-T cell therapy consisted of 88 patients. The baseline clinical characteristics are summarized in Table 1. The minor CD19 allele L174 was detected in 51 individuals (58% of the cohort). The median age at time of CAR-T cell therapy was 67 years. A male predominance was present in the CD19 V174 group (70% vs. 45%, *p* = 0.029). All patients were diagnosed r/r DLBCL. The proportion of de novo versus transformed lymphoma was without distribution difference (*p* = 0.66). The majority in both groups presented disease stage IV at the time of diagnosis according to Ann-Arbor classification system. All patients received at least one treatment line prior to CAR-T cell therapy. More patients in the L174 group (0% vs. 14%, *p* = 0.019) received CAR-T as second line therapy. However, no difference was observed in the distribution of 2, 3, or >3 treatment lines before CAR-T between the two groups. In both groups, the majority of the patients received second line chemotherapy followed by ASCT prior to CAR-T as third line therapy (76% vs. 65%, *p* = 0.36). All patients were treated with R-CHOP chemotherapy regimen as first line therapy. There was no difference in the proportion of patients receiving radiotherapy or hematopoietic stem cell transplantation in previous treatment lines.

### 3.3. Disease Features and CAR-T Cell Treatment

The treatment characteristics are summarized in Table 2. The International Prognostic Index (IPI) at the time before CAR-T cell therapy was available in 74 patients. An IPI ≥ 3 was observed in the majority of patients in both groups (68% vs. 65%, *p* = 0.82). r/r DLBCL before CAR-T cell infusion was present as progressive disease in 57% of the CD19 V174 group and in 47% of the L174 group (*p* = 0.27), which was confirmed by PET-CT. Bridging chemotherapy was administered in 51% of the CD19 V174 patients and in 33% of the L174 group (*p* = 0.4). All patients received lymphodepleting chemotherapy with Fludarabine and Cyclophosphamide days −5 to −3 before CAR-T cell infusion. The majority of the patients were treated with Tisagenlecleucel (Kymriah©), 30% with Axigabtagene ciloleucel (Yescarta©), and 7% with Lisocabtagene Maraleucel (Breyanzi©, JCAR017). The time between lymphocyte apheresis and CAR-T cell infusion differed between the groups, with median times of 41 and 38 days, respectively (*p* = 0.0275).

Cytokine release syndrome (CRS) at all grades after CAR-T cell infusion was present in both groups (78% vs. 78%, *p* > 0.99). CAR-T-related encephalopathy syndrome (CRES) after CAR-T cell infusion occurred at all grades in 43% vs. 29% (*p* = 0.26) and CRES occurred at higher grades (III and IV) in the CD19 V174 group, at 25% vs. 16% (*p* = 0.14).

The peak value of measured CAR-T level was detected after a median duration of 10 vs. 9 days (*p* = 0.11). The peak values of CAR-T persistence (4212 vs. 5432 copies per µg cfDNA, *p* = 0.86) did not differ between the two groups.

### 3.4. Treatment Outcome, Univariate Analysis

In this retrospective observational study, we analyzed a cohort of 88 patients with r/r DLBCL comparing two genetically variant groups with single nucleotide polymorphism in the CD19 gene encoding leucine or valine at amino acid position 174 (CD19 L174 or V174). The two subgroups were comparable regarding baseline clinical characteristics (Table 1) and median follow-up times (24.3 vs. 22.9 months, *p* = 0.96). 

The outcome of CAR-T cell treatment was analyzed for the entire cohort and for the subgroups of CD19 L174 versus V174 (Figure 1, Table 3). Patients carrying the germline variant CD19 L174 allele had a better treatment outcome (Figure 1A,B). Transformed DLBCL had a better outcome than de novo DLBCL, with a median OS of 7.9 months for de novo and 3 years for transformed DLBCL (*p* = 0.011) (Figure 1C,D). Younger patients had a better treatment outcome, with a median OS of four years in patients up to 65 years of age and only 7 months in patients over 65 years of age at the start of treatment (*p* = 0.002) (Figure 1E,F).

The median PFS was 6.1 months in the CD19 V174 group and 22.1 months in the L174 group (*p* = 0.0592). The median OS was 8 and 37 months, respectively (*p* = 0.1136). A total of 32% vs. 14% (V174 vs. L174) had primary refractory disease (*p* = 0.0367). In patients achieving complete or partial response, 35% vs. 31% relapsed after a median duration of 6.1 vs. 3.7 months (*p* = 0.4). Death occurred in 57% vs. 43% after a median of 2.8 vs. 7.7 months after CAR-T cell treatment (*p* = 0.6). After a median follow up of 20 months, the complete response rate (CRR) after CAR-T cell therapy was 30% in the CD19 V174 group and 51% in the L174 group (*p* = 0.0525). The overall response rate (ORR) was 65% vs. 76% respectively, (*p* = 0.24). The overall best response was reached within 3 months in 70% vs. 61% (*p* = 0.37). The distribution of CAR-T persistence was correlated with clinical outcomes after CAR-T cell therapy [11]. Circulating CAR-T DNA was equally persistent in the CD19 L174 patients as in the CD19 V174 patients treated with FMC63-anti-CD19-CAR-T cell therapy (Figure 2).

### 3.5. Treatment Outcome, Multivariate Analysis

The standard approach of applying univariate tests on individual response variables has the advantage of simplicity of interpretation, but it fails to account for covariance/correlation in the data. In contrast, multivariate statistical techniques might more adequately capture the multi-dimensional pathophysiological pattern and therefore provide increased sensitivity to detect treatment effects. The multivariate analysis included parameters of possible impact on clinical outcome, including age at treatment start (>65 vs. ≤65 years), sex, disease status (transformed versus de novo DLBCL), type of CAR-T cell product (Tisa-cel versus Axi-cel), and response (CR within six months versus no CR).

The multivariate analysis supported the trend observed in the univariate analysis for the CD19 minor allele L174 to be a prognostic indicator for treatment outcome with HR = 0.64 at a *p*-value of 0.1172 (Table 4). Additionally, age at treatment start was a prognostic factor associated with outcome after CAR-T cell therapy, as previously reported [31,32]. Early CR was the most significant favorable indicator for treatment outcome, as described in clinical trials [4,32]. Disease status was predictive for treatment outcome as median OS times were significantly longer in transformed versus de novo DLBCL. Transformed lymphoma was previously associated with a favorable response to CAR-T cell treatment in DLBCL patients [9]. Finally, the type of CAR-T cell product also affected outcome, with longer OS times in patients treated with Axi-cel (Yescarta©). A matched comparison study supported the higher efficacy and toxicity of Axi-cel (Yescarta©) compared to Tisa-cel (Kymriah©) in the third or more treatment line for R/R DLBCL [33].

## 4. Discussion

CD19-directed CAR-T cell therapy has become a widely established treatment option in r/r DLBCL patients relapsing after first or second line therapy [1,2,3,4,5,34,35]. Even though response rates to CAR-T cell therapy are remarkably high, a significant number of r/r DLBCL patients eventually relapse after achieving complete response [7,8]. Loss of CD19 antigen appears to be important in relapses of B-ALL in younger patients [17,18], but the role of CD19 status in relapsed DLBCL in adult patients is controversial [24]. Somatic mutations in exons 1–6 of CD19 can facilitate alternative splicing, resulting in loss or modulation of the epitope, possibly masking the target cell to CAR-T cells. These mechanisms have been investigated in pediatric patients with B-ALL relapsing after CAR-T [19,20,21,22].

Germline CD19 variation may be a prognostic marker for CAR-T cell treatment outcome in DLBCL. In this study, we present evidence for a superior clinical outcome to CAR-T cell therapy in patients carrying the single nucleotide polymorphism rs2904880 encoding the germline variant L174 of the CD19 antigen. Significant differences were observed for primary refractory disease at a rate of 32% in CD19 V174 carriers compared to 14% in CD19 L174 carriers (*p* = 0.0367). Response rates were reduced in CD19 V174 carriers compared to CD19 L174 (CRR: 30% vs. 51%, *p* = 0.0525; ORR: 65% vs. 76%, *p* = 0.2438). The median survival times were reduced in the CD19 V174 group, with 6 and 8 months (PFS, OS) compared to 22 and 37 months (PFS, OS) in the CD19 L174 carriers (PFS: *p* = 0.0592; OS: *p* = 0.1136).

The correlation of the CD19 L174 variant and a favorable outcome in the observed DLBCL cohort is in accordance with a recent study, where lymphoma cells with a somatic mutation encoding CD19 V174 were reported to show insufficient response to FMC63-CAR19-T cell therapy compared to CD19 L174 lymphoma cells [26]. Fine mapping of the CD19 extracellular domain revealed that epitopes for three antibodies FMC63, 4G7, and 3B10 partially overlapped near residues W140–G150 and P200-P203, with distinct per-site and per-mutation tolerances [36]. L174 may subtly change the presentation of the CD19 epitope to the FMC63 antibody, thereby reducing the binding affinity (Figure 3). CD19 L174 has been described as the first minor histocompatibility antigen on hematopoietic cells presented by HLA class II (HLA-DQA1*05/B1*02) molecules to CD4 T cells encoded by a single nucleotide polymorphism (SNP) in the CD19 gene [32]. At a minor allele frequency (MAF) of 0.3, the corresponding CD19 variant frequencies are L/L = 9%, L/V = 42%, and V/V = 49%, with 51% prevalence of the L variant. The germline variant CD19 L174 genotype frequency of 45–55% mirrors the complete response rate of 43–54% in FMC63-CAR-T-treated DLBCL patients. Interestingly, CD19 L174 may be a clinical variant as rs2904880 has been described in genome-wide association studies as a heritable risk for Parkinson’s disease [37].

As a potential prognostic indicator for treatment outcome in FMC63-CAR-T cell therapy, the prevalence of the CD19 minor allele L174 is of interest. With a global minor allele frequency of 0.29, half of the patients would be expected to carry the L174 allele. The minor allele frequency, however, may vary from 0.04 to 0.4 depending on the subpopulation (Asian 0.08, African 0.1, European 0.31, Latin American 0.4). At a minor allele frequency of 0.1, only 20% of the patients would be expected to carry the L174 allele. In the treatment of DLBCL patients homozygous for CD19 V174, a CAR-T construct with a different single-chain variable fragment (scFv) may be more effective, as reported for 21D4 CAR-T cells [26].

## 5. Conclusions

In this retrospective observational study, we investigated the treatment outcome related to the single nucleotide polymorphism rs2904880 encoding leucine or valine at amino acid position 174 (L174V) on the CD19 antigen in a DLBCL patient cohort receiving FMC63-anti-CD19-CAR-T therapy. Our data suggest that the single nucleotide polymorphism in CD19 influences the treatment outcome in FMC63-anti-CD19-CAR-T cell therapy, and that the CD19 minor allele L174 predicts a favorable treatment outcome. To confirm the impact of CD19 polymorphism in FMC63-CAR-T response, a larger retrospective study of FMC63-treated patients is required. Moreover, to improve treatment outcome in CD19 V174 homozygous DLBCL patients, the development of novel anti-CD19 CAR-T constructs is imperative.

## Figures and Tables

**Figure 1 cancers-15-03058-f001:**
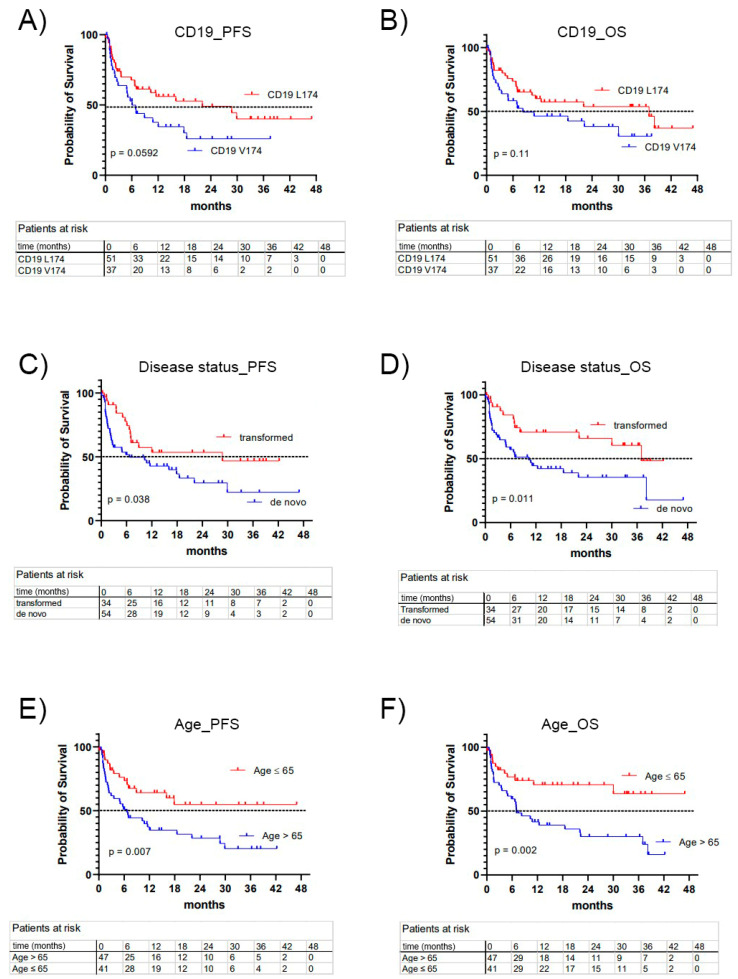
Clinical outcome in DLBCL patients receiving FMC63-CAR-T cell therapy. Survival times were analyzed for progression-free survival (PFS) and overall survival (OS). (**A**) PFS according to CD19 gene polymorphism rs2904880 encoding CD19 V174 or L174. (**B**) OS in subgroups CD19 V174 and L174. (**C**) PFS according to disease status, transformed versus de novo DLBCL. (**D**) OS according to disease status, transformed versus de novo DLBCL. (**E**) PFS according to age at treatment start, age ≤ 65 years vs. age > 65 years. (**F**) OS according to age at treatment start, age ≤ 65 years vs. age > 65 years.

**Figure 2 cancers-15-03058-f002:**
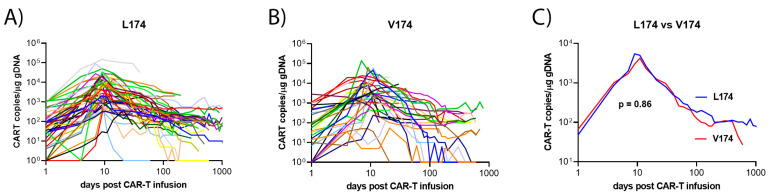
CAR-T cell dynamics in the plasma of DLBCL patients treated with FMC63-anti-CD19-CAR-T cell therapy. CAR-T copies per microgram genomic DNA in the plasma of CD19 L174 patients (*n* = 48) treated with FMC63-anti-CD19-CAR-T cell therapy (**A**); CD19 V174 patients (*n* = 34) treated with FMC63-anti-CD19-CAR-T cell therapy (**B**); moving median plot of CAR-T plasma levels in CD19 L174 versus CD19 V174 carriers (**C**).

**Figure 3 cancers-15-03058-f003:**
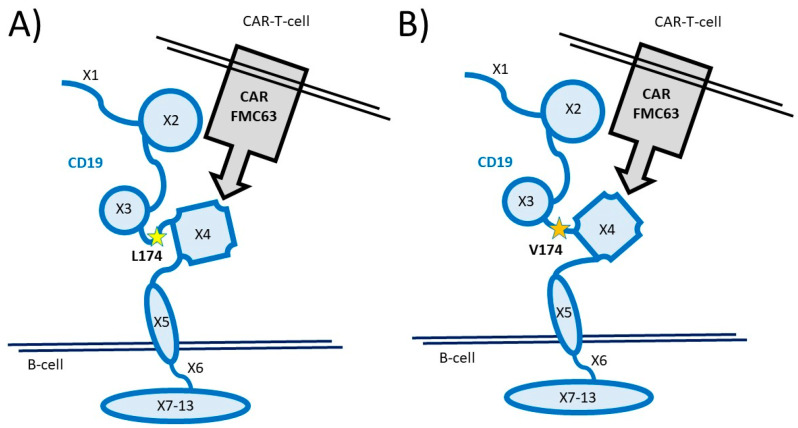
Schematic presentation of the two germline variants of the CD19 antigen encoded by single nucleotide polymorphism (snp) rs2904880. Extracellular protein domains are encoded by exons 1 to 5, intracellular domains by exons 6 to 13. The anti-CD19-FMC63 expressed on CAR-T cells may exhibit a greater binding affinity to the CD19 L174 variant (**A**) and reduced affinity to the CD19 V174 variant (**B**). At a minor allele frequency (MAF) of 0.3, the corresponding CD19 variant frequencies are L/L = 9%, L/V = 42%, and V/V = 49%, with 51% prevalence of the L variant.

**Table 1 cancers-15-03058-t001:** Baseline clinical parameters.

Parameter	All Patients	CD19 V174	CD19 L174	*p*-Value *
Patients *n* (%)	88 (100)	37 (42)	51 (58)	
Male to female ratio	49:39 (1.3)	26:11 (2.4)	23:28 (0.8)	0.029
Age at the time of CAR-T cell therapy, median (range)	67 (35–82)	68 (42–82)	66 (35–79)	0.36
Initial diagnosis				
DLBCL, *n* (%)	88 (100)			
de novo DLBCL, *n* (%)	54 (61)	24 (65)	30 (59)	0.66
transformed DLBCL, *n* (%)	34 (39)	13 (35)	21 (41)	0.66
Transformed from:				
FL, *n* (%)	24 (27)	9 (24)	15 (15)	>0.99
CLL, *n* (%)	5(6)	1 (3)	4 (8)	0.63
MCL, *n* (%)	3(3)	1 (3)	2 (4)	>0.99
other, *n* (%)	2 (2)	2 (5)	0	0.14
Stage at initial diagnosis				
I, *n* (%)	2 (2)	1 (3)	1 (2)	>0.99
II, *n* (%)	18 (20)	7 (19)	11 (22)	0.79
III, *n* (%)	17 (19)	8 (22)	9 (18)	0.78
IV, *n* (%)	49 (55)	21 (57)	28 (55)	>0.99
Unknown	2 (3)	0	2 (4)	
Number of treatment lines before CAR-T cell therapy, *n* (%)				
1	7 (8)	0	7 (14)	0.019
2	63 (72)	28 (76)	33 (65)	0.35
3	13 (15)	6 (16)	8 (16)	>0.99
>3	5 (6)	2 (5)	3 (6)	>0.99
Previous radiotherapy	16 (18)	5 (14)	11 (22)	0.41
Previous SCT	44 (49)	21 (57)	23 (45)	0.39
Autologous SCT	43 (48)	20 (54)	23 (45)	0.52
Allogeneic SCT	1 (1)	1 (3)	0	0.42

CAR: chimeric antigen receptor; DLBCL: diffuse large B-cell lymphoma; FL: follicular lymphoma; CLL: chronic lymphocytic leukemia; MCL: mantle cell lymphoma; SCT: stem cell transplantation; * univariate analysis.

**Table 2 cancers-15-03058-t002:** Clinical characteristics and details of CAR-T cell treatments.

Parameter	All Patients	CD19 V174	CD19 L174	*p*-Value *
Patients *n* (%)	*n* = 88 (100)	*n* = 37 (42)	*n* = 51 (58)	
IPI before CAR-T cell therapy				
1	3 (5)	2 (5)	1 (2)	0.57
2	13 (14)	6 (16)	7 (14)	0.77
3	36 (41)	12 (32)	24 (47)	0.19
4	20 (23)	11 (30)	9 (18)	0.31
5	2 (2)	2 (5)	0 (0)	0.17
Nd	14 (16)	4 (11)	10 (20)	
Remission Status at CAR-T infusion				
CR	3 (3)	0	3 (6)	0.26
PR	22 (25)	11 (30)	11 (22)	0.46
SD	18 (20)	5 (14)	13 (25)	0.19
PD	45 (51)	21 (57)	24 (47)	0.27
Bridging chemotherapy	36 (41)	19 (51)	17 (33)	0.40
Bridging radiotherapy	11(13)	7 (19)	7 (14)	0.56
Lymphodepleting chemotherapy with Fludarabidine, Cyclophosphamide	88 (100)			
LDH before CAR-T (U/L), median (range)	458 (164–3949)	457 (164–2355)	472 (189–3949)	0.49
Median time between lymphapheresis and CAR-T cell infusion, days (range)	40 (13–170)	41 (26–170)	38 (13–112)	0.027
CAR-T cell product				
Kymriah©	56 (64)	22 (59)	34 (67)	0.59
Yescarta©	26 (30)	11 (30)	15 (29)	>0.99
Breyanzi©	6 (7)	4 (11)	2 (4)	0.24
Cytokine release syndrome	69 (78)	29 (78)	40 (78)	>0.99
Grade 1	41 (47)	17 (46)	27 (53)	0.66
Grade2	21 (24)	11 (30)	10 (20)	0.32
Grade 3	3 (4)	1 (3)	2 (4)	>0.99
Grade 4	4 (5)	0	1 (2)	>0.99
CAR-T-related encephalopathy syndrome	31 (35)	16 (43)	15 (29)	0.26
Grade 1	9 (10)	4 (11)	5 (10)	>0.99
Grade 2	5 (7)	2 (5)	3 (6)	>0.99
Grade 3	13 (13)	8 (22)	5 (10)	0.14
Grade 4	4 (5)	1 (3)	3 (6)	0.64
Peak value CAR-T copies per µg cfDNA, median (range)	4860 (37–218,384)	4212 (37–139,656)	5432 (193–218,384)	0.86
Time to peak value, median (range), days	9 (2–46)	10 (2–46)	9 (5–41)	0.11

IPI: international prognostic index; CR: complete response; PR: partial response; SD: stable disease; PD: progressive disease; LDH: lactate dehydrogenase; cfDNA: cell-free DNA. * univariate analysis.

**Table 3 cancers-15-03058-t003:** Clinical outcome after CAR-T cell therapy.

Parameter	CD19 V174	CD19 L174	*p*-Value *
	*n* = 37 (42)	*n* = 51 (58)	
Best response after CAR-T cell therapy			
CR, *n* (%)	11 (30)	26 (51)	0.05
PR, *n* (%)	13 (35)	13 (25)	0.35
SD, *n* (%)	5 (14)	5 (10)	0.73
PD, *n* (%)	8 (22)	6 (12)	0.25
Time to best response (months), *n* (%)			
1, *n* (%)	26 (70)	31 (61)	0.38
3, *n* (%)	8 (22)	15 (29)	0.47
6, *n* (%)	1 (3)	1 (2)	>0.99
12, *n* (%)	1 (3)	4 (8)	0.39
18, *n* (%)	0 (0)	0 (0)	>0.99
24, *n* (%)	1 (3)	1 (2)	>0.99
30, *n* (%)	0 (0)	1 (2)	>0.99
Overall response (CR or PR) within 6 months, *n* (%)	24 (65)	39 (76)	0.24
CR after 1 year, *n* (%)	13 (35)	25 (49)	0.27
CR at last follow-up, *n* (%)	13 (35)	27 (53)	0.13
Primary refractory disease	12 (32)	7 (14)	0.036
Relapse after achieving CR, *n* (%)	7 (19)	11 (22)	0.79
Relapse treatment	13 (35)	16 (31)	0.82
Pharmacotherapy	8 (22)	11 (22)	
Radiotherapy	5 (14)	6 (12)	
Median time to relapse, months (range)	6.1 (2.2–17.9)	3.7 (1.4–28.8)	0.39
Death	21 (57)	22 (43)	0.28
Median time to death, months (range)	2.77 (0.5–22.27)	4.73 (0.1–38.2)	0.60
Median follow up time (months)	24.33	22.85	0.96
Median PFS (months)	6.1	22.10	0.059
Median OS (months)	8.27	37	0.113

PFS: progression-free survival; OS: overall survival; CR: complete response; PR: partial response; SD: stable disease; * univariate analysis.

**Table 4 cancers-15-03058-t004:** Clinical outcome after CAR-T cell therapy, multivariate analysis.

	*PFS*		*OS*	
Predictors	HR (95% CI)	*p*-Value *	HR (95% CI)	*p*-Value *
CD19 L174	0.63	0.1172	0.64	0.1772
Age > 65	1.8	0.0699	2.53	0.0101
Male sex	1.09	0.7864	1.29	0.4234
Transformed DLBCL	0.56	0.0852	0.40	0.0139
Yescarta© (vs. Kymriah©)	0.65	0.1801	0.57	0.1125
CR within 6 months	0.17	<0.0001	0.13	<0.0001

CR: complete remission; HR: hazard ratio, * multivariate analysis.

## Data Availability

Data available on request due to restrictions, privacy and ethics.

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
