# Peer review of "Clinical Impact of Single Nucleotide Polymorphism in CD-19 on Treatment Outcome in FMC63-CAR-T Cell Therapy"

_cancers, 2023, doi:10.3390/cancers15113058_

Round 1
Reviewer 1 Report
The article entitled 'Clinical Impact of Single Nucleotide Polymorphism in CD-19 on Treatment Outcome in FMC63-CAR-T Cell Therapy’ is very interesting, and presents pivotal data related to SNP’s in CD19 patients. The article needs to include some more information.
The authors could improve the review article. Below are a few comments.
1. In Figure 2A, Can the authors confirm if all the responders and non responders data was presented in the figure, they should mention the number (n) as It is unclear how many are non responders and we don’t see CART copies increasing by Day 10. Not clear if they are included in the analysis or not?
2. In Figure 2B, Similar to the above comment Can the authors confirm if all the responders and non responders data was presented in the figure, they should mention the number (n) It is unclear how the data of the patients were presented, as there are a few non responders and we don’t see CART copies increasing by Day 10. Are these included in the analysis or not?
3. Looks like both the L174 vs V174 show similar profile, can the authors include how the looks like in normal patients treated with CART product apart from these specific patients.
4. Whether the authors investigated other side effects apart from Cytokine release syndrome such as B-cell dysplasia or other common symptoms.
5. Any patients have been treated with IL-6 therapy, or had seen elevated levels of IL6 ?
6. Can the others compare across CART therapies such as Axicabtagene, Ciloleucel (Axi-cel) and Tisagenlecleucel (Tisa-cel).
7. There is a typo error in the front page for the word 'analysis' ?
8. Please include the article by Cameron J. Turtle et al, CD19 CAR–T cells of defined CD4+:CD8+ composition in adult B cell ALL patients
Reviewer 2 Report
CD19-targeting CAR T-cell therapy is effective for treating patients with relapsed or refractory diffuse large B-cell lymphoma with a CRR of 50%. It is reported that the antigen expression, the T cell status, and immune suppression in patients are critical for CAR-T efficacy. However, whether single nucleotide polymorphism of CD19 can affect the CAR-T therapy is not clear. Seipel et. al reported that SNP of rs2904880 has clinical impact on FMC63-anti-CD19 CAR-T therapy. This research is interesting and timely. It provides new insights for therapeutic outcome prediction and developing novel CD19-targeting CAR-T therapy.
Some minors need to be clarified:
According to Table 2, there are three patients in CR status at CAR-T infusion. Why these patients do the CAR-T treatment?
Table 3, “months” is missing in the row of “Median follow up time”
Line 60, Reference 16 should be move to the end of the sentence “the target can 58 be modified in order to evade cytotoxicity induced by the immune system of the body or 59 by immunologic therapies.”
Line 79, “the light chain fraction” should be “the single chain fragments variable (scFv) fraction” or similar description.
Line 230, FAM63 should be FMC63.
Line 387, the format of reference 19 is incorrect.
no comments
